# Reactions of NO₃ with Aromatic Aldehydes: Gas-Phase Kinetics and Insights into the Mechanism of the Reaction

Yangang Ren,[1] Li Zhou,[1,2] Abdelwahid Mellouki,[1,3] * Véronique Daële,[1] Mahmoud Idir,[1] Steven S. Brown,[4,5] Branko Ruscic,[6] Robert S. Paton,[7] Max R. McGillen,[1] and Akkihebbal R. Ravishankara[1,7,8,9] *

1. Institut de Combustion Aérothermique, Réactivité et Environnement, Centre National de la Recherche Scientifique (ICARE-CNRS), Observatoire des Sciences de l'Univers en région Centre (OSUC), CS 50060, 45071 cedex02 Orléans, France
2. Present address: College of Architecture and Environment, Sichuan University, Chengdu, 610065, China.
3. Environment Research Institute, School of Environmental Science and Engineering, Shandong University, Qingdao 266237, China
4. NOAA, Chemical Sciences Laboratory, Boulder, CO USA
5. Department of Chemistry, University of Colorado, Boulder, CO USA
6. Chemical Sciences and Engineering Division, Argonne National Laboratory, Lemont, IL 60439, USA
7. Department of Chemistry, Colorado State University, Fort Collins, CO USA
8. Department of Atmospheric Science, Colorado State University, Fort Collins, CO USA
9. Le Studium Institute for Advanced Studies, Orléans, France

*: Address correspondence to:

mellouki@cnrs-orleans.fr or A.R.Ravishankara@Colostate.edu.

**Abstract:** Rate coefficients for the reaction of $NO_3$ radicals with a series of aromatic aldehydes were measured in a 7300-liter simulation chamber at ambient temperature and pressure by relative and absolute methods. The rate coefficients for benzaldehyde (BA), ortho-tolualdehyde (O-TA), meta-tolualdehyde (M-TA), para-tolualdehyde (P-TA), 2,4-dimethyl benzaldehyde (2,4-DMBA), 2,5-dimethyl benzaldehyde (2,5-DMBA) and 3,5-dimethyl benzaldehyde (3,5-DMBA) were: $k_1 = 2.6 \pm 0.3$, $k_2 = 8.7 \pm 0.8$, $k_3 = 4.9 \pm 0.5$, $k_4 = 4.9 \pm 0.4$, $k_5 = 15.1 \pm 1.3$, $k_6 = 12.8 \pm 1.2$ and $k_7 = 6.2 \pm 0.6$, respectively, in the units of $10^{-15}$ cm$^3$ molecule$^{-1}$ s$^{-1}$ at 298$\pm$2 K. The rate coefficient $k_{13}$ for the reaction of the $NO_3$ radical with deuterated benzaldehyde (benzaldehyde-d$_1$) was found to be half that of $k_1$. The end product of the reaction in an excess of $NO_2$ was measured to be $C_6H_5C(O)O_2NO_2$. Theoretical calculations of aldehydic bond energies and reaction pathways indicate that $NO_3$ radical reacts primarily with aromatic aldehydes through the abstraction of an aldehydic hydrogen atom. The atmospheric implications of the measured rate coefficients are briefly discussed.

Keywords: rate coefficient, $NO_3$ radical, aromatic aldehydes, mechanism

# 1. Introduction

Aromatic aldehydes are a family of organic compounds emitted into the atmosphere from anthropogenic and pyrogenic sources. For example, benzaldehyde (BA) has been detected in the incomplete combustion of fuels (Legreid et al., 2007). Methylated benzaldehydes (ortho, meta, para-tolualdehydes) are present in wood smoke and biomass burning plumes (Koss et al., 2018). Benzaldehyde also has extensive industrial usage in perfume, soap, food and drink production, and as a solvent for oils and resins (OECD, 2002). Bezaldehyde and ortho/meta/para-tolualdehyde could also be formed from the atmospheric degradation of aromatic hydrocarbons, although their yields in the atmosphere are expected to be small (Calvert et al., 2002;Obermeyer et al., 2009).

It is known that the tropospheric removal of aromatic aldehydes occurs through photolysis (Clifford et al., 2011) and their reactions with the OH radical. During daytime, photolysis and reaction with OH dominate the atmospheric loss of the aromatic aldehydes. (We assume that the aromatic aldehydes photodissociate akin to aliphatic aldehydes). Yet, oxidation of aromatic aldehydes via their reaction with the nitrate radical, $NO_3$, may be important in NOx-rich locations at night since both the aromatic aldehydes and the nitrate radical can arise from anthropogenic emissions and pyrogenic source. To the best of our knowledge, there are no studies that have measured the formation of secondary organic aerosol (SOA) from the title reaction in either a chamber or in the atmosphere. One could suspect that aromatic aldehydes may be degraded at night with consequences for ozone and SOA formation. The reactions of aromatic aldehydes with $NO_3$ likely lead to acyl peroxy nitrates (APNs, often referred to as PANs) in the NOx-rich regions with characteristically high $NO_3$ abundances. The APNs would then transport nitrogen oxides and the aromatic moieties from the polluted to cleaner parts of the globe (Wayne, 2000) with consequences for ozone production and particle formation away from the polluted regions. Therefore, quantifying the kinetics and understanding the mechanism of the reaction of $NO_3$ with aromatic aldehydes is needed.

In this study, the rate coefficients $k_1$-$k_7$ at 298 K for the reactions of $NO_3$ radical with the following seven aromatic aldehydes were measured:

$$\text{benzaldehyde (BA)} + NO_3 \; \rightarrow \; \text{products;} \qquad k_1 \tag{1}$$

$$\text{ortho - Tolualdehyde (O-TA)} + NO_3 \; \rightarrow \; \text{products;} \qquad k_2 \tag{2}$$

$$\text{meta - Tolualdehyde (M-TA)} + NO_3 \; \rightarrow \; \text{products;} \qquad k_3 \tag{3}$$

$$\text{para - Tolualdehyde (P-TA)} + NO_3 \; \rightarrow \; \text{products;} \qquad k_4 \tag{4}$$

$$\text{2,4 - dimethyl benzaldehyde (2,4-DMBA)} + NO_3 \; \rightarrow \; \text{products;} \qquad k_5 \tag{5}$$

$$\text{2,5 - dimethyl benzaldehyde (2,5-DMBA)} + NO_3 \; \rightarrow \; \text{products ;} \qquad k_6 \tag{6}$$

$$\text{3,5 - dimethyl benzaldehyde (3,5-DMBA)} + NO_3 \rightarrow \text{products;} \qquad k_7 \tag{7}$$

The rate coefficient for Reaction (1) has been previously reported by Carter et al. (1981), Atkinson et al. (1991), Clifford et al. (2005) and Bossmeyer et al. (2006). The rate coefficients for Reactions 2-4 were reported only by Clifford et al. (2005). There are no previous reports on $k_5$-$k_7$, to the best of our knowledge. The results from earlier studies are compared with our data in the section of Results and Discussions.

We report the rate coefficients at 298 K for the above reactions, and we have determined the stable products formed in Reaction (1). We also have attempted to elucidate the mechanism of these reactions via studies of isotopic substitution, quantum chemistry calculations, and an examination of the linear free energy relationship.

## 2. Experimental setup and procedures

The kinetics and products were studied in a 7300-liter indoor simulation chamber described in detail previously (Zhou et al., 2017); therefore, it is described only briefly here. The chamber was made of Teflon foil. Two Teflon fans located inside the chamber rapidly mixed the contents of the chamber within about 30 s. Purified air was used as a bath gas and to flush the chamber to clean it in between experiments. Upon flushing, the levels of $NO_2$ and $O_3$ were less than 50 pptv (detection limit of our instruments). A proton transfer reaction time-of-flight mass spectrometer (PTR-TOF-MS) and a Fourier transform infrared spectrometer (FT-IR, Nicolet 5700) coupled to a White-type multipass cell (143 m optical path length) were employed to monitor the organic compounds in the chamber. The white cell was located within the chamber. An inlet situated in the center of the chamber fed the PTR-ToF-MS. The masses in PTR-ToF-MS and IR bands in the FTIR used to measure the aldehydes are listed in the SI (Table S1). The PTR-ToF-MS and/or FTIR signals were measured by expanding a known mass of liquid aldehyde in to the chamber to determine hydrocarbons concentrations. Calibration plots for the quantification of the aromatic aldehydes and the reference gases were constructed using these measurements. They are shown in Figure S1 in the SI. The concentrations of $NO_3$ and $N_2O_5$ were determined using a cavity ring down spectrometer (CRDS). The $NO_3$ radical was detected using its strong 662 nm absorption. The sum of $NO_3$ and $N_2O_5$ were detected simultaneously by thermally dissociating $N_2O_5$ in a second channel. The time resolution of CRDS was 1s, and detection limits for $NO_3$ and $N_2O_5$ were approximately 0.5 pptv ($1.23 \times 10^7$ molecule cm$^{-3}$ at atmospheric pressure) for a 5 s integration (Brown et al., 2002; Dubé et al., 2006). $NO_3$ radicals were generated from the thermal decomposition of $N_2O_5$ in the chamber.

### 2.1 Rate coefficient measurements

**Relative Method**: The temporal profiles of a reference compound and the aromatic aldehyde

of interest were monitored simultaneously using the PTR-TOF-MS and/or FTIR in the presence

of $NO_3$ radical.

$NO_3$ + aromatic aldehydes → products; $k_{1-7}$

$NO_3$ + references → products; $k_{ref}$

We accounted for the dilution of the chamber necessitated by the addition of air to keep the pressure constant while we continually withdrew the chamber's contents for analysis. Similarly, we also accounted for the depletion of the hydrocarbons due to the loss to the walls. The two processes together are represented as a first-order loss process with a rate coefficient $k_d$:

aromatic aldehydes/references → wall loss/dilution; $k_d$

Above, $k_{1-7}$ and $k_{ref}$ are the rate coefficient for the reaction of $NO_3$ with studied aromatic aldehydes (BA, O-TA, M-TA, P-TA, 2,4-DMBA, 2,5-DMBA, and 3,5-DMBA) and the

reference compound (methyl methacrylate, MMA), respectively. The dilution rate coefficient was measured by adding a small amount of unreactive $SF_6$ into the chamber and monitoring its temporal decay. We quantified the rate coefficient (ranging from $5.5 \times 10^{-7}$ $s^{-1}$ to $1.6 \times 10^{-6}$ $s^{-1}$) for the removal of the aldehydes and reference compound due to loss on the walls by monitoring their decay in the absence of the $NO_3$ reactant

The typical experimental procedure for relative rate measurements consisted of a sequence of three steps: (1) $SF_6$ was added at the beginning of the experiment and monitored to measure the dilution rate. During this time, as noted above, air was continually added to make up for the loss due to the withdrawal for analysis; (2) VOCs (aromatic aldehydes, reference compounds) were introduced into the chamber and monitored for roughly 30 minutes to obtain $k_d$; and (3)

$N_2O_5$ was then continually introduced to the chamber using pure air as a carrier gas, and the consumption of VOCs was monitored using PTR-TOF-MS for 1-2 hours. The typical initial VOC concentrations were $(0.9-4.5) \times 10^{12}$ molecule $cm^{-3}$ (Table S2). Typical $NO_3$ concentrations were $(0.25-2.5) \times 10^9$ molecule $cm^{-3}$.

Assuming that the aromatic aldehydes and reference compounds were lost by reaction with

$NO_3$ radical and dilution/loss to the walls, it can be shown that:

$$\ln(\frac{[aro]_0}{[aro]_t}) - k_d \times t = (\frac{k_{1-7\_RR}}{k_{ref}}) \times [\ln(\frac{[ref]_0}{[ref]_t}) - k_d \times t \,] \qquad \text{I}$$

Where $[aro]_0$ and $[aro]_t$, and $[ref]_t$ and $[ref]_0$, are the corresponding concentrations of aromatic

aldehydes and reference compound at the initial time and time t, respectively. According to

Equation I, plots of $\{\ln(\frac{[aro]_0}{[aro]_t}) - k_d \times t\}$ against $\{\ln(\frac{[ref]_0}{[ref]_t}) - k_d \times t\}$ are straight lines of slope

equal to $k_{1-7}/k_{ref}$, with zero intercepts.

**Absolute method**: Experiments were conducted in the same 7300-liter simulation chamber described in the previous section. The temporal profiles of $NO_3$ and $N_2O_5$ were measured using CRDS in an excess of each aromatic aldehyde (Zhou et al., 2017). They were simultaneously fit to a reaction scheme (see below) to extract the reaction rate coefficient as described by Zhou et al. (2017). The typical experimental procedure consisted of the following steps: (1) $N_2O_5$ was introduced with pure air into the chamber, and the temporal profiles of $NO_3$ and $N_2O_5$ concentrations were measured to determine the rate coefficient for loss of $NO_3$ and $N_2O_5$ to the walls, reactions with impurities, and dilution; (2) The aromatic aldehyde of interest was introduced while continually measuring the temporal profiles of $NO_3$ and $N_2O_5$; and (3) $SF_6$ was introduced to determine the dilution rate when needed.

A box model was used to integrate Reactions (1-11) to obtain the rate coefficients of $NO_3$ radical reaction with aromatic aldehydes. The obtained temporal profiles of $NO_3$ and $N_2O_5$ concentrations were fit to the observed profiles using a non-linear least squares algorithm.

$$NO_3 + NO_2 \rightarrow N_2O_5 \; ; k_8 \tag{8}$$

$$N_2O_5 \rightarrow NO_3 + NO_2 \; ; k_9 \tag{9}$$

$$NO_3 \rightarrow wall\ loss \; ; k_{10} \tag{10}$$

$$N_2O_5 \rightarrow wall\ loss \; ; k_{11} \tag{11}$$

$$NO_3 + aromatic\ aldehydes \rightarrow products \; ; k_{1-7} \tag{1-7}$$

In the algorithm, the sum of squares the difference between calculated and measured values of both the $NO_3$ and $N_2O_5$ were minimized while varying the input parameters. In the absence of the aldehydes, the input parameter was just $k_d$. The value of $k_d$ obtained was then held constant when subsequently fitting the second set of temporal profiles where the rate coefficient for reactions (1)-(7) was varied. The other input parameters included the measured initial concentration of $NO_3$ ($[NO_3]_0$), $N_2O_5$ ($[N_2O_5]_0$), and the initial aromatic aldehydes ($[aro]_0$) as well as the temperature-dependent values of $k_8$ and $k_9$ (to define the equilibrium constant ($K_{eq}$)). We used the equilibrium constant for the reaction: $NO_3 + NO_2 \rightleftharpoons N_2O_5$ recommended by Burkholder et al. (2015). The $NO_2$ concentration was calculated through the equilibrium constant and measured concentrations of $NO_3$ and $N_2O_5$ throughout the course of the reactions; i.e., $[NO_2]=[N_2O_5]/([NO_3] \times K_{eq})$ and used in the fits. Such calculated temporal profiles of $NO_3$ and $N_2O_5$ were fit to the measured temporal profiles at various concentrations of the aldehydes. As shown in Table S3, the concentration of aromatic aldehyde was always 50-1500 times higher

than that of $NO_3$ in the chamber at all times. Thus $NO_3$ loss was essentially first order in its concentration.

## 2.2 Chemicals

The aromatic aldehydes were purchased from Sigma-Aldrich. The stated purities of these chemicals were: benzaldehyde ($\geq$ 99.5%), O-TA (97%), M-TA (97%), P-TA ($\geq$ 97%), 2,4-DMBA ($\geq$ 90%), 2,5-DMBA (99%), and 3,5-DMBA (97%). Methyl methacrylate was bought from TCI. The chemical purity of benzaldehyde-α-d1 was 99%, while its isotopic purity was 98%. All the aldehydes (in liquid form) were further purified by repeated freeze-pump-thaw

cycles before use. Substantial concentrations of reactive impurities were not detected in these samples based on the PTR-TOF-MS/FTIR measurements. A mixture of $NO_2$ and $O_3$ was flowed into a 1-liter bulb to generate $N_2O_5$ through Reactions 12 and 8.

$$NO_2 + O_3 \rightarrow NO_3 + O_2 ; k_{12} \tag{12}$$

$$NO_3 + NO_2 \rightarrow N_2O_5 ; k_8 \tag{8}$$

The $N_2O_5$ crystals were collected in a cold trap (190 K) and purified by trap-to-trap distillation in a mixture of $O_2/O_3$. $N_2O_5$ was stored in a cold trap maintained at 190 K.

# 3. Results and Discussion

## 3.1 Rate coefficients determination of $NO_3$ reaction with a series of aromatic aldehydes

In the study, we measured the rates coefficients $k_1$-$k_7$ using a relative method and an absolute method. They are outlined separately below for ease of presentation.

### 3.1.1 Relative method

Methyl methacrylate, MMA, was used as the reference compound in this work because the rate

coefficient at 298 K for its reaction with $NO_3$ radical has been measured in the same chamber using the same method as here (Zhou et al., 2017) to be $k_{MMA}$=(2.98±0.35)×10$^{-15}$ cm$^3$ molecule$^{-1}$ s$^{-1}$. We acknowledge that there are other recommendations that are slightly higher than the value we used. One can always renormalize the rate coefficient for any specific value. The experimental conditions and associated parameters are shown in Table S2.

Figure 1 shows plots of $\{\ln(\frac{[aro]_0}{[aro]_t}) - k_d \times t\}$ against $\{\ln(\frac{[ref]_0}{[ref]_t}) - k_d \times t\}$ for the seven aromatic aldehydes. Each plot is linear with an intercept of zero within the uncertainties. The $k_d \times t$ term is relatively small (3.6-7.6%) compared to the ratio $\{\frac{k_d \times t}{\ln(\frac{[aro]_0}{[aro]_t}) \text{ or } \ln(\frac{[ref]_0}{[ref]_t})}\}$, and slight variations in its value do not affect the accuracy of the measured rate coefficients. Figure 1 also shows that the measured values of the rate coefficient ratios are reproducible. The ratios of rate coefficients,

$\frac{k_{1,7}}{k_{ref}}$, were calculated from these plots using the algorithm of Brauers and Finlayson-Pitts (1997), which takes into account errors in both the abscissa and ordinate. These errors were estimated from the uncertainty of the concentration calculated from the calibration plots generated before the kinetics runs. The noted errors for $\frac{k_{1,7}}{k_{ref}}$ are twice the standard deviation in the least-squares fits multiplied by a factor to account for the limited number of measurements using the Student

$t$-distribution.

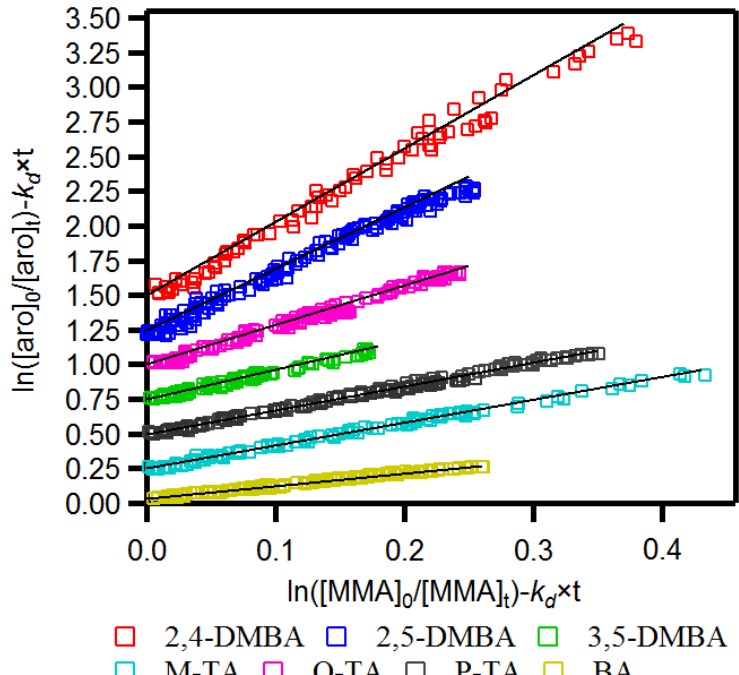

**Figure 1.** Plots of relative kinetic data obtained from the reaction of aromatic aldehydes (aro) with NO$_3$ radical using methyl methacrylate (MMA) as the reference. M-TA, P-TA, 3,5-DMBA,

O-TA, 2,5-DMBA and 2,4-DMBA were shifted 0.25, 0.5, 0.75, 1.0, 1.25, and 1.5, respectively, for clarity.

The relative rate coefficients of the studied aromatic aldehydes, termed k$_{RR}$, are shown in Table S2. The estimated uncertainties for k$_{RR}$ were the sum of the precision of our measurements

(noted above) and the quoted uncertainties in the rate coefficient for the reference reaction according to the expression:

$$\sigma(k_{RR}) = \frac{k}{k_{ref}} \times k_{ref} \sqrt{\left[\frac{\sigma_{k_{ref}}}{k_{ref}}\right]^2 + \left[\frac{error}{\frac{k}{k_{ref}}}\right]^2}.$$

**3.1.2 Absolute method**

The method used to analyze the temporal profiles of $NO_3$ and $N_2O_5$ to obtain $k_1$-$k_7$ has been described by Zhou et al. (2017;2019). Figure 2 shows an example of the temporal profiles of $NO_3$ and $N_2O_5$ for Reaction (1) from which the rate coefficient $k_1$ was derived. Similar analyses yielded $k_2$-$k_7$ (Figure S2-7).

       Panel (a) of Figure 2 shows the temporal profile of $NO_3$ and $N_2O_5$ concentrations plotted
on a logarithmic scale. The concentrations decrease exponentially (the lines are linear in the log-space), and the decay rates are lower in the absence of the reactant aldehyde than in its presence. The rate coefficients for the loss of $NO_3$ and $N_2O_5$ to the walls and dilution, $k_{10}$ and $k_{11}$, were determined to be in the ranges $(3.7\text{-}7.1)\times10^{-3}$ $s^{-1}$ and $(3.2\text{-}12.0)\times10^{-4}$ $s^{-1}$, respectively. As Zhou et al. (2017), we cannot merely take the slopes of decay of $NO_3$ with time to calculate
the rate coefficients $k_1$-$k_7$ since $NO_3$ and $N_2O_5$ are coupled through their equilibrium. The equilibration is maintained throughout the course of Reactions (1)-(7). To account for this situation, we fit the profiles of both $NO_3$ and $N_2O_5$ to a reaction scheme, as described previously.

       Panels (b) and (c) of Figure 2 show the $NO_3$ and $N_2O_5$ temporal profiles on a linear scale in the absence and the presence of the reactant aldehyde. The two panels also show the fits of
the data to the mechanism that included only Reactions (1)-(11). The fits are acceptable but with larger variations at longer reaction times. We have to consider the contributions to $NO_3$ and $N_2O_5$ losses due to the reactions of $NO_3$ with the products of the Reactions (1)-(7). Values of the rate coefficients derived from such fits included all the potential reactions that can contribute to the removal of $NO_3$, and they are shown in the SI (Table S4). When the secondary
reactions with the products were included, the fits were better at the longer reaction times, and they are shown in Panel (d) of Figure 2. The residuals of the fits are shown at the bottoms of Panel (c) and (d); they clearly show the improvement in the fits and the lack of a trend with reaction time. Larger deviations are to be expected in Panel (c) at longer reaction times as the reaction products build up. The rate coefficients $k_1$-$k_7$ calculated using the reaction scheme
shown in Table S4 are taken to be those measured by the direct method. As expected, the rate coefficients calculated by including the contributions of the secondary reactions were slightly less than those without the secondary reaction contributions (see Figure S8). The differences were, on average, about 5%.

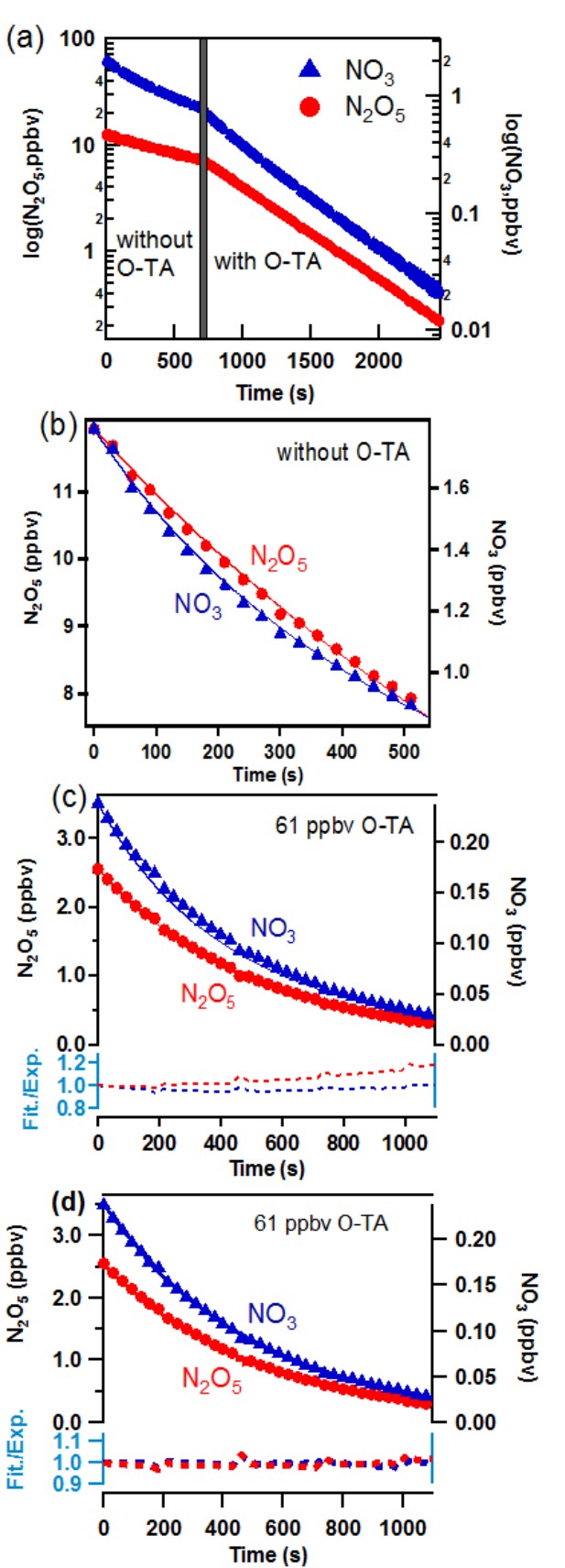


**Figure 2** Observed (points) and simulated (lines) profiles of $N_2O_5$ (left axis) and $NO_3$ (right axis) as a function of time. Mixing ratios of o-tolualdehyde are shown in each Panel. Panel (a): the temporal profiles on a log scale. Panel (b): Temporal profiles on a linear scale and the fits

to the data as discussed in the text to determine the wall loss of $N_2O_5$ and $NO_3$ during the experiment. Panel (c): Fits of simulations not including the subsequent secondary reactions of $NO_3$ with the products of the initial reaction. The dash lines at the bottom show the ratio of simulated lines to measured data points (Fit./Exp.) for $NO_3$ (blue) and $N_2O_5$ (red). Panel (d): Fits of simulations including the subsequent secondary reactions of $NO_3$ with the initial reaction products (shown in Table S4). The dash lines at the bottom show the ratio of simulated lines to measured data points (Fit./Exp.) for $NO_3$ (blue) and $N_2O_5$ (red). The improvements in the fits are clear in Panel (d).

The results of our measured values for $k_{1-7}$ are summarized in Table S3. The quoted errors of the rate coefficient from each experiment are at the 95% confidence level based on the precision of the fits; they were typically less than 7%. The weighted averages of results from multiple experiments were calculated, and then the influence of the small number of measurements was accounted for by using a Student $t$-distribution table. We added the estimated systematic uncertainties to the precision in quadrature, assuming that they are uncorrelated. Contributions to estimated systematic errors included: (1) The systematic errors of -8/+11% and -9/+12%, respectively, in the measurements of $NO_3$ and $N_2O_5$ (Zhou et al., 2019); (2) The uncertainty of around 10% in the rate coefficients used in the reaction schemes shown in Table S4; and (3) The estimated uncertainty of 7% in the concentration of aromatic aldehydes. This includes the uncertainties in the calibration and spectral analysis. All the noted uncertainties are at the 95% confidence level, assuming a Gaussian error distribution.

It is important to note that we need the absolute concentrations of $NO_3$ and $N_2O_5$ even though the reaction was first order in $NO_3$ due to the strong coupling between the concentrations of $NO_3$ and $N_2O_5$ via an equilibrium. The presence of the aldehydes would influence the temporal profiles of both $N_2O_5$ and $NO_3$. Here we are attributing the entire change to the reaction of $NO_3$ with the aldehydes. The validity of this assumption is shown by the measured rate coefficients being independent of the ratio of $[NO_3]/[N_2O_5]$. This ratio was changed simply by changing $[NO_2]$ that shifts the equilibrium concentrations of $NO_3$ and $N_2O_5$.

### 3.1.3 Comparison of Rate coefficients obtained from absolute and relative methods

The rate coefficients for the reactions of $NO_3$ with seven different aromatic aldehydes, $k_1$-$k_7$, measured using the absolute and relative methods are summarized in Table 1. The rate coefficient values from the two methods are in good agreement with each other. The differences are less than 10%, except for $k_1$, which differs by 18%.

As shown in Table 2, the final rate coefficients of 7 aromatic aldehydes reaction with $NO_3$ radical were derived from the weighted average of the absolute and the relative rate methods, using the equation discussed above (Eq. S2 and Eq. S3).

**Table 1.** Rate coefficients $k_1$-$k_7$ measured in the study using two different methods.

| Aromatic aldehydes | rate coefficient ($10^{-15}$ cm$^3$ molecule$^{-1}$ s$^{-1}$) | | ratio ($k_{RR}/k_{AR}$) | $k_{final}$[a] ($10^{-15}$ cm$^3$ molecule$^{-1}$ s$^{-1}$) |
|---|---|---|---|---|
| | $k_{RR}$ | $k_{AR}$ | | |
| BA | 2.8±0.4 | 2.3±0.4 | 1.18 | 2.5±0.3 |
| O-TA | 8.5±1.0 | 9.1±1.2 | 0.93 | 8.7±0.8 |
| M-TA | 5.1±0.6 | 4.7±0.6 | 1.07 | 4.9±0.5 |
| P-TA | 5.0±0.6 | 4.8±0.6 | 1.05 | 4.9±0.4 |
| 2,4-DMBA | 15.5±1.8 | 14.6±2.0 | 1.06 | 15.1±1.3 |
| 2,5-DMBA | 13.4±1.6 | 12.2±1.7 | 1.09 | 12.8±1.2 |
| 3,5-DMBA | 6.4±0.8 | 6.1±0.8 | 1.05 | 6.2±0.6 |

[a] weighted average values from three measurements and their error calculated using Eq.S2 and Eq.S3

**Table 2** Experimental results of this study compared with those from the literature for the reactions of NO$_3$ radical with benzaldehyde (BA), o-tolualdehyde (O-TA), m-tolualdehyde (M-TA), p-tolualdehyde (P-TA), 2,4-dimethylbenzaldehyde (2,4-DMBA), 2,5-dimethylbenzaldehyde (2,5-DMBA) and 3,5-dimethylbenzaldehyde (3,5-DMBA).

| VOC | k ($\times10^{-15}$ cm$^3$ molecule$^{-1}$ s$^{-1}$) | T (K) | Technique | Reference compound | Ref. ($\times10^{-15}$ cm$^3$ molecule$^{-1}$ s$^{-1}$) | Ref. |
|---|---|---|---|---|---|---|
| BA | <8 | 300±1 | relative | propene | 7.90 | Carter et al. 1981 |
| | 1.1±0.3 | 296±2 | relative | propene | 4.07 | Atkinson et al. 1984 |
| | 2.6±0.1 | 296±2 | relative | propene | 9.50 | Atkinson et al. 1991 |
| | 4.2±0.2 | 295±2 | relative | tetrahydrofuran | 4.90 | Clifford et al. 2005 |
| | 4.5±0.3 | 295±2 | relative | n-propyl ether | 4.90 | Clifford et al. 2005 |
| | 2.2±0.6 | 301±3 | absolute | | | Bossmeyer et al. 2006 |
| | 2.8±1.0 | 298±1 | relative | MMA | 2.98±0.35 | this work |
| | 2.3±0.4 | 298±1 | absolute | | | this work |
| | **2.6±0.3** [a] | 298±2 | | | | **recommended** |
| O-TA | 9.3±0.3 | 295±2 | relative | tetrahydrofuran | 4.90 | Clifford et al. 2005 |
| | 10.3±0.4 | 295±2 | relative | n-propyl ether | 4.90 | Clifford et al. 2005 |
| | 8.5±1.6 | 298±1 | relative | MMA | 2.98±0.35 | this work |
| | 9.1±1.3 | 298±1 | absolute | | | this work |
| | **8.7±0.8** [b] | 298±1 | | | | **recommended** |
| M-TA | 9.4±0.4 | 295±2 | relative | tetrahydrofuran | 4.90 | Clifford et al. 2005 |
| | 9.6±0.4 | 295±2 | relative | n-propyl ether | 4.90 | Clifford et al. 2005 |
| | 4.9±1.2 | 298±1 | relative | MMA | 2.98±0.35 | this work |
| | 4.6±0.7 | 298±1 | absolute | | | this work |
| | **4.9±0.5** [b] | | | | | **recommended** |
| P-TA | 8.4±0.7 | 295±2 | relative | tetrahydrofuran | 4.90 | Clifford et al. 2005 |
| | 10.2±0.4 | 295±2 | relative | n-propyl ether | 4.90 | Clifford et al. 2005 |
| | 5.0±1.2 | 297±1 | relative | MMA | 2.98±0.35 | this work |
| | 4.8±0.7 | 297±1 | absolute | | | this work |
| | **4.9±0.4** [b] | | | | | **recommended** |
| 2,4-DMBA | 15.8±2.2 | 297±1 | relative | MMA | 2.98±0.35 | this work |
| | 14.6±2.0 | 297±1 | absolute | | | this work |
| | **15.1±1.3** [b] | | | | | **recommended** |
| 2,5-DMBA | 13.2±1.9 | 297±1 | relative | MMA | 2.98±0.35 | this work |
| | 12.2±1.7 | 297±1 | absolute | | | this work |

| | | | | | | |
|---|---|---|---|---|---|---|
| | **12.8±1.2 [b]** | | | | | **recommended** |
| 3,5-DMBA | 6.2±1.3 | 297±1 | relative | MMA | 2.98±0.35 | this work |
| | 6.1±0.9 | 297±1 | absolute | | | this work |
| | **6.2±0.6 [b]** | | | | | **recommended** |

[a] weighted average of Atkinson (1991), Bossmeyer et al. (2006) and this work including results of relative and absolute method.
[b] values shown in Table 1.

### 3.1.4 Comparison with the literature for NO$_3$ radical kinetic with aromatic aldehydes

Table 2 summarizes the rate coefficients measured in this work with data from the literature for the reactions of the NO$_3$ radical with aromatic aldehyde, BA, O-TA, M-TA, P-TA, 2,4-DMBA, 2,5-DMBA, and 3,5-DMBA. As shown in Table 2, the rate coefficient for BA has been reported by five studies. Three of them (Atkinson et al., 1984;Carter et al., 1981;Clifford et al., 2005) used the relative method with different reference compounds. Atkinson et al. (1991) corrected the values from their earlier report, and they are used for the comparison. Bossmeyer et al.,(2006) measured both NO$_3$ and BA using differential optical absorption spectroscopy (DOAS) in their chamber. They measured the loss of BA in a known (measured continuously) NO$_3$ concentration and fitted BA's measured temporal profile to obtain $k_1$. Calvert et al.,(2011) recommended $k_7$ to be 4.0 ×10$^{-15}$ cm$^3$ molecule$^{-1}$ s$^{-1}$ with a 30% uncertainty based on these studies. However, our value from absolute and relative methods using methyl methacrylate as reference reaction (its rate coefficient has been determined using absolute method in our previous study (Zhou et al., 2017)) is in good agreement with Atkinson (1991) and Bossmeyer et al. (2006). Hence, we suggest that the weighted average based on these three studies, 2.6±0.3 ×10$^{-15}$ cm$^3$ molecule$^{-1}$ s$^{-1}$ at 298±2 K, is a reliable value. The rate coefficients for ortho/meta/para-tolualdehyde have only been studied by Clifford et al. (2005) who reported them to be (9.8±0.4), (9.5±0.4) and (9.5±0.7) ×10$^{-15}$ cm$^3$ molecule$^{-1}$ s$^{-1}$, respectively. In this work, $k_1$-$k_7$ were determined relative to MMA as the reference as well as using an absolute method to be: (8.5±1.0 & 9.1±1.2), (5.1±0.6 & 4.7±0.6) and (5.0±0.6 & 4.8±0.6) ×10$^{-15}$ cm$^3$ molecule$^{-1}$ s$^{-1}$, respectively. This work agrees best with the value of Clifford *et al*. (2005) for ortho-tolualdehyde, but those of $k_3$ and $k_4$ are smaller than those of Clifford et al. (2005). The reasons for these discrepancies are unclear, but the excellent agreement (<7% difference) between the two techniques presented here gives us confidence in our determinations. This work provides the first experimental determinations of the rate coefficients for NO$_3$ reactions with 2,4-DMBA, 2,5-DMBA, and 3,5-DMBA, where the two complementary methods agree well. The recommended rate coefficients of the weighted average of relative method and absolute method are shown in Table 2.

### 3.2 Mechanisms of the reactions

### 3.2.1 Products investigation from the reaction of benzaldehyde with NO₃

The stable products formed in Reaction (1) were investigated at 298±2 K and 760 Torr in the same 7300-liter simulation chamber. Benzaldehyde, 0.9-1.4×10¹³ molecule cm⁻³ (as shown in Table S5), was introduced into the chamber, and its removal was measured for 2 hours to obtain the wall loss rate coefficient. Then, roughly 1.1×10¹²-1.5×10¹⁴ molecule cm⁻³ of N₂O₅ was introduced. Stable products formed in the chamber were identified and quantified (when possible) using the PTR-TOF-MS and FTIR.

Two stable products, C₆H₅C(O)O₂NO₂ (benzaldehyde-PAN; BAPAN) and C₆H₅ONO₂, were detected and measured. The former was detected using both PTR-TOF-MS (m/z 184.024 and its fragment m/z 105.034) and FTIR (965-1005 cm⁻¹ centered at 989 cm⁻¹) and the latter using only PTR-TOF-MS. Since we do not have a sample of BAPAN, we could not quantify the yield of this product using PTR-TOF-MS. However, Caralp *et al.* (1999) have reported the IR band strengths for BAPAN. Using their reported band strength, we could quantify BAPAN to be 80±10% of the benzaldehyde that was removed via reaction, where the quoted uncertainty is the precision in the fit at the 2 σ level. When we account for the uncertainties in the absorption cross sections of BAPAN reported by Caralp et al. (1999) (~20%) and the uncertainties in the concentration of initial benzaldehyde concentration (~10%), and add the precision of the measurements, we conclude that the yield of BAPAN is 80±22%. We assume that the uncertainties are uncorrelated and hence added them in quadrature. The obtained BAPAN amounts are shown in Figure 3 as the function of benzaldehyde consumption; the details are shown in Table S6 in the SI. We could not quantify C₆H₅ONO₂ because of the lack of a standard. Assuming that the ion-molecule reaction rate coefficients for proton transfer to BAPAN and C₆H₅ONO₂ are similar, we estimate that the yield of C₆H₅ONO₂ is smaller than that of BAPAN. These two products are expected if the reaction proceeds via H atom abstraction, most of the peroxy radical reacts with NO₂ (Platz et al., 1998) as denoted by the reaction scheme A:

(A)

A fraction of the peroxy radicals reacts with itself (or other peroxy radicals) to make the phenoxy radicals, which ultimately leads to a nitrate, according to the mechanism B:

(B)

Unfortunately, we could not reduce the concentration of NO₂ sufficiently to suppress pathway B completely. A small fraction of the C₆H₅C(O)O₂ would also react with NO₃ but would still

yield the $C_6H_5C(O)O$ radicals. Also, any reactions of phenyl radical with $NO_2$ can be neglected because of the large abundance of $O_2$ that will quickly convert it to $C_6H_5O_2$ radical. Numerical modeling of the reaction sequence shown in the SI (Table S4) suggests that the yield of BAPAN is more than 95% under our experimental conditions. Based on these results, we suggest the yield of BAPAN in our reaction system to be essentially 1 and that we detect the nitrate because of the high sensitivity for its detection in our PTR-TOF-MS.

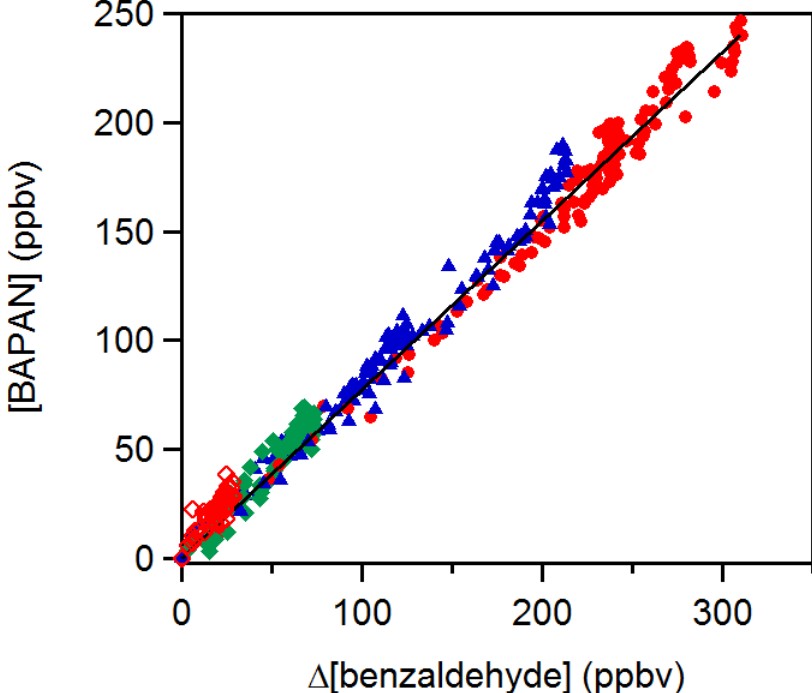

**Figure 3.** A plot of benzaldehyde PAN ($C_6H_5C(O)O_2NO_2$; BAPAN) concentrations measured by FTIR as a function of depletion of benzaldehyde. The slope of the plot gives the yield of the BAPAN in the NOx-rich environment to be roughly 80±20%. Note: the different colors/markers presented the repeated experiments with different initial $NO_2$ concentrations as: red filled circles: $[NO_2]_0$= 1.4-2.9 ×$10^{12}$ molecule cm$^{-3}$ molecule cm$^{-3}$, blue triangle: $[NO_2]_0$= ~2.8 ×$10^{13}$ molecule cm$^{-3}$, green square : $[NO_2]_0$= ~1.4×$10^{14}$ molecule cm$^{-3}$, red open diamond: $[NO_2]_0$= ~3.3 ×$10^{14}$ molecule cm$^{-3}$, filled red circles.

**3.2.1 Kinetic Isotope Effect in the reaction**

To further examine the mechanism of $NO_3$ reactions with the aromatic aldehydes, we measured the rate coefficient for the reaction of $NO_3$ radical with benzaldehyde-α-d1 ($C_6H_5CDO$):

$$C_6H_5C(O)D + NO_3 \rightarrow \text{products} ; k_{13} \qquad (13)$$

As shown in Figure 4, $k_{13}$ is half that of $k_1$, i.e., $k_1/k_{13}$ is 1.92. A factor of 2 decrease in the rate coefficient going from benzaldehyde to benzaldehyde-α-d1 is consistent with a primary kinetic

isotope effect (KIE), suggesting that abstraction of the aldehydic H atom occurs in the rate-limiting step of this reaction pathway (See calculated KIE below).

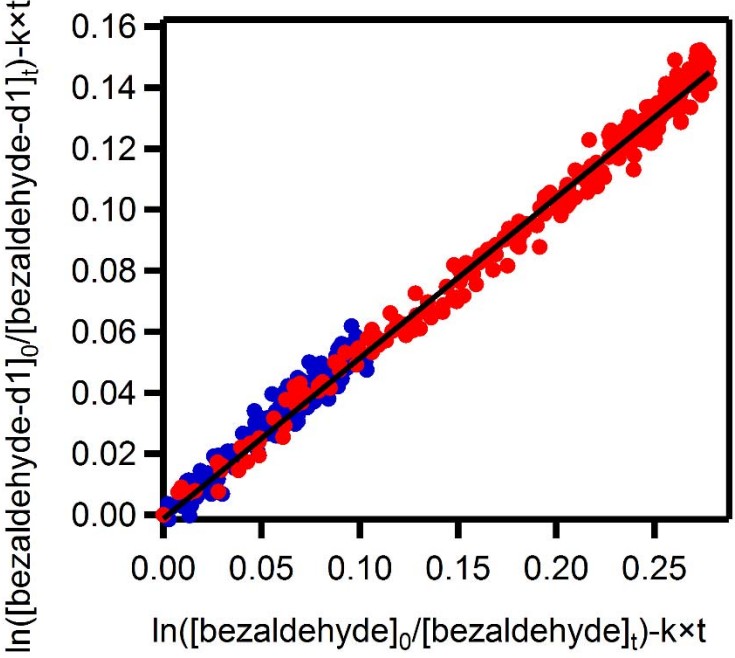

**Figure 4.** The plot of the logarithm (base e) of the ratio of initial benzaldehyde-D1 concentration to those at other reaction times against the logarithm (base e) of the ratio of the initial benzaldehyde concentration relative to those at different times. Both benzaldehyde and benzaldehyde-D1 were competing for the same pool of $NO_3$ radicals. The slope of the plot is 0.53±0.01. The red and blue circles present two experiments with different concentrations of $NO_3$ radicals.

### 3.2.2 Reaction pathway

Linear free energy relationships comparing the reactivities of OH and $NO_3$ with a series of organic compounds are shown in the SI (Figure S9). Such correlations have been demonstrated in the past by Clifford et al. (2005). We have added our measured values of $k_1$-$k_7$ to the plot. This plot suggests that the linear free energy relationship is consistent with $NO_3$ radicals adding to the seven aromatic aldehydes studied here. Even the reaction of OH radicals with aliphatic aldehydes is known to proceed by forming a pre-reaction complex.

Figure 5 shows the rate coefficients for the reaction of the $NO_3$ radical with the seven aldehydes studied here (also presented in Table 2), along with those with benzene and toluene ($<3\times10^{-17}$ and $<6.6\times10^{-17}$ cm$^3$ molecule$^{-1}$ s$^{-1}$) (Calvert et al., 2011). Benzene and toluene do not react with $NO_3$ to measurable extents; such reactions may be expected if they were proceeding via electrophilic addition. Since it is not sufficiently reactive to abstract an H atom from either

the ring or the methyl group, the rate coefficient is very slow, if not zero. Therefore, the

observed reaction rate coefficients suggest that the reaction proceeds via an H-atom abstraction from the –CHO group (Clifford et al., 2005;Wayne et al., 1991). The rate coefficient for the reaction of $NO_3$ with benzaldehyde, $(2.6\pm0.3)\times10^{-15}$ $cm^3$ molecule$^{-1}$ s$^{-1}$, is similar to that with acetaldehyde $(2.7\pm0.5)\times10^{-15}$ $cm^3$ molecule$^{-1}$ s$^{-1}$, as shown in Figure 5. The mechanism for the reaction of $NO_3$ with acetaldehyde is believed to be H-atom abstraction from the –CHO group

after the formation of a pre-reaction complex.

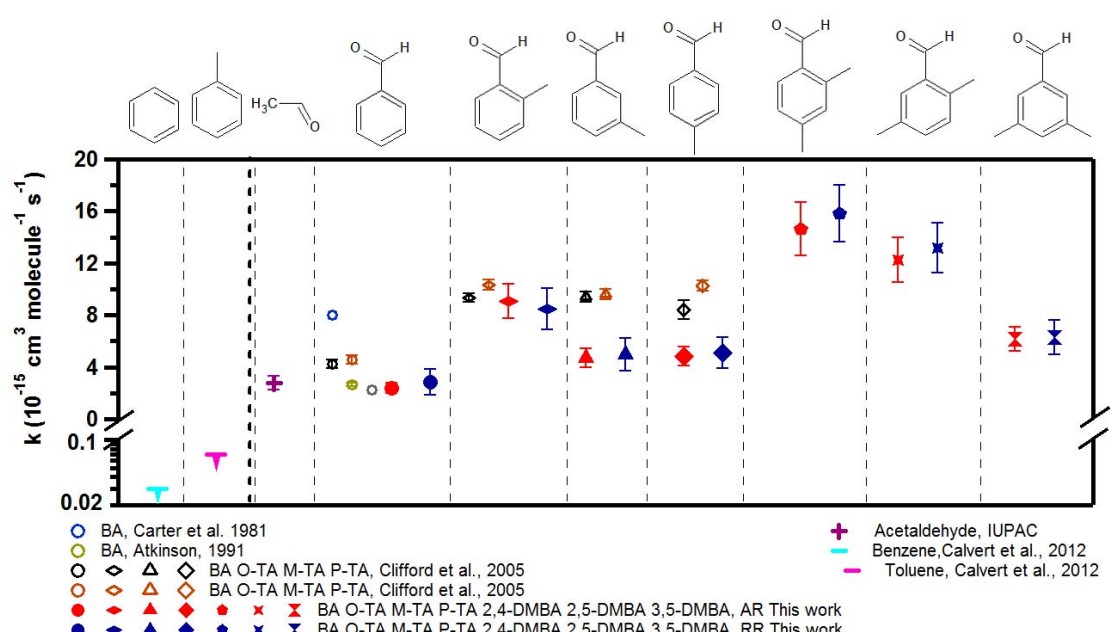

**Figure 5** The measured rate coefficients for the reactions of $NO_3$ radical with the seven

aldehydes studied here. The results of previously reported rate coefficients are also shown (values are presented in Table 2). The rate coefficient for acetaldehyde of $k_{(CH3CHO)}=(2.7\pm0.5)\times10^{-15}$ $cm^3$ molecule$^{-1}$ s$^{-1}$ is from IUPAC recommendations. The rate coefficients for benzene, $k_{(benzene)}<3\times10^{-17}$ $cm^3$ molecule$^{-1}$ s$^{-1}$, and for toluene, $k_{(toluene)}\leq6.6\times10^{-17}$ $cm^3$ molecule$^{-1}$ s$^{-1}$ are from Calvert et al., 2011. 88.4 90.2


As shown in Figure 5 (also presented in Table 2), this work finds a higher rate coefficient for the reactions of $NO_3$ with ortho/meta/para-tolualdehydes than that with benzaldehyde. This enhanced reactivity with ring substitution by electron-donating groups suggests an electrophilic role for $NO_3$ in the reaction. Clifford et al. (2005) noted that the direct influence of the electron-

donation to the ring by the $CH_3$ group is effectively canceled out by the electron-withdrawing effect of the –CHO group as an explanation for their measured rate coefficients being the same for all the tolualdehydes. The trends we observe for substituting methyl groups in the tolualdehydes and the demethylated aldehydes could suggest electrophilic addition. Of course,

the initial addition has to lead to the abstraction of the aldehydic H atom, as shown by the

products and the kinetic isotope effect. Alternatively, the reactivity trend could simply be due to the C-H bond energy changes in the aldehydic group upon methyl substitution.

The C-H bond dissociation enthalpies (BDEs) of benzaldehyde and the three tolualdehydes have been obtained from Active Thermochemical Tables (ATcT). As opposed to traditional sequential thermochemistry, ATcT obtains enthalpies of formation by constructing, statistically

analyzing, and solving a global thermochemical network (TN), which is formed by including experimental and theoretical determinations pertinent to the chemical species that are included in the network, as explained in more details elsewhere (Ruscic et al., 2004;Ruscic et al., 2005).

The results presented here are based on the most current ATcT TN (ver. 1.122x), obtained by further expanding prior (Bross et al., 2019;Zaleski et al., 2021) versions (1.122r and 1.122v) by

including species of interest in the present study, as well as other ongoing investigations. The current TN incorporates ~2,350 species, interconnected by ~29,000 experimental and theoretical determinations.

Typically, the insertion of a new chemical species in the TN begins by linking the new species to the already existing species via a provisional skeleton of theoretical isodesmic reactions computed

using a standard set of mid-level composite calculations carried in-house, and currently consisting of W1 (Martin and de Oliveira, 1999;Parthiban and Martin, 2001), CBS-APNO (Ochterski et al., 1996), G4 (Curtiss et al., 2007), G3X (Curtiss et al., 2000), and CBS-QB3 (Montgomery et al., 1999, 2000). This is subsequently complemented by experimental and theoretical determinations from the literature and, when possible, additional state-of-the-art high-level composite calculations that can

deliver sub-kJ mol$^{-1}$ accuracies. As the number and accuracy of additional determinations grow, the dependence of the final result on the determinations spanning the initial skeleton diminishes and ultimately vanishes.

However, while the enthalpies of formation of gas-phase benzaldehyde and the three tolualdehydes are linked in the TN to their condensed phases, for which there are some

thermochemically-relevant experimental data in the literature, the reported BDEs rely heavily on the results of mid-level composite methods, given the paucity of literature on experimental or theoretical data involving the benzoyl and toluyl radicals and the prohibitively high cost of state-of-the-art high-level composite calculations for this size of species. Analogously, our attempts to expand the TN with dimethyl benzaldehydes and their related radicals were frustrated by a virtual

absence of experimental and worthwhile theoretical data in the literature, combined with the high cost of theoretical calculations even using mid-level composite methods.

The ATcT BDEs of the aldehydic C-H in benzaldehyde and the three tolualdehydes are given at 298.15 K and 0 K in Table S6, together with the corresponding enthalpies of formation of the parents and the related radicals. While only the 298.15 K BDEs are discussed below, the

corresponding 0 K BDEs (a.k.a. $D_0$ values) are also given in the same table and are, as expected, approximately 6.3 kJ mol$^{-1}$ (or ~2.5 RT) lower. The ATcT uncertainties provided in Table S6

correspond to 95% confidence intervals, following the standard in thermochemistry (Ruscic, 2014;Ruscic and Bross, 2019), and were obtained by using the full ATcT covariance matrix. Consequently, when the enthalpy of formation of the radical is highly correlated to that of the parent, as happens to be true in benzaldehyde and tolualdehydes, the uncertainty of the resulting BDE is perceptibly lower than the uncertainty that would be obtained by manually propagating the uncertainties of the individual enthalpies of formation in quadrature, since the latter summation assumes zero covariances.

The ATcT BDE of the aldehydic C-H in benzaldehyde is $BDE_{298}(C_6H_5C(O)-H) = 380.10 \pm 0.84$ kJ mol$^{-1}$. This is noticeably higher (by $6.77 \pm 0.84$ kJ mol$^{-1}$) than the corresponding BDE of the prototypical aldehyde – acetaldehyde – for which the current version of the ATcT TN produces $BDE_{298}(CH_3C(O)-H) = 373.37 \pm 0.29$ kJ mol$^{-1}$ (essentially unchanged from the web-accessible value in the earlier ATcT TN ver. 1.122p (Rusic and Bross, 2020).

Of particular relevance here is the fact that the aldehydic C-H BDEs in meta- and para-tolualdehyde, $BDE_{298}(m\text{-}CH_3C_6H_4C(O)-H) = 379.93 \pm 1.13$ kJ mol$^{-1}$ and $BDE_{298}(p\text{-}CH_3C_6H_4C(O)-H) = 379.85 \pm 1.13$ kJ mol$^{-1}$, are indistinguishable (within the uncertainty) from each other and when compared to the BDE in benzaldehyde. However, in ortho-tolualdehyde, the corresponding BDE is consistently lower essentially by 4.2 kJ mol$^{-1}$, $BDE_{298}(o\text{-}CH_3C_6H_4C(O)-H) = 376.09 \pm 1.13$ kJ mol$^{-1}$, lower by $3.85 \pm 1.00$ kJ mol$^{-1}$ than that of meta-tolualdehyde, $3.76 \pm 1.00$ kJ mol$^{-1}$ than that in para-tolualdehyde, and $4.01 \pm 0.92$ kJ mol$^{-1}$ than that in benzaldehyde. Therefore, the likely origin of the reactivity differences is simply due to the bond enthalpies in the abstraction of aldehydic H atoms.

Based on the discussion above, it appears that the preponderance of the evidence is consistent with the abstraction of the aldehydic H atom. However, could such an abstraction reaction start via the addition of NO$_3$ to the ring followed by abstraction? To examine this possibility, we carried out quantum mechanical calculations of the reaction pathways in Reaction (1). Stationary points on the potential energy surface (PES) were optimized with the BH&HLYP density functional and 6-311G(d,p) basis set in Gaussian 16 (Becke, 1993;Frisch et al., 2016), except for the NO$_3$ radical, for which the MP2-D3h geometry was used. This level of theory and empirical treatment of NO$_3$ follows Boyd's computational studies with smaller aldehydes, XCHO (X = H, F, Cl, Me) (Mora-Diez and Boyd, 2002). Single point energies were evaluated at the DLPNO-CCSD(T)/cc-pVTZ level of theory with tight SCF convergence and TightPNO cutoffs in Orca (Neese, 2012;Riplinger and Neese, 2013;Riplinger et al., 2013). Relative energetics with this basis set are essentially converged, since the effect of using cc-pVQZ on the activation barrier is less than 1 kJ mol$^{-1}$. The calculated PES is shown in Figure 6. Computed enthalpic and Gibbs energy changes include unscaled vibrational zero-point

energies and translational, rotational, and vibrational contributions at 298.15 K (Luchini et al., 2020).

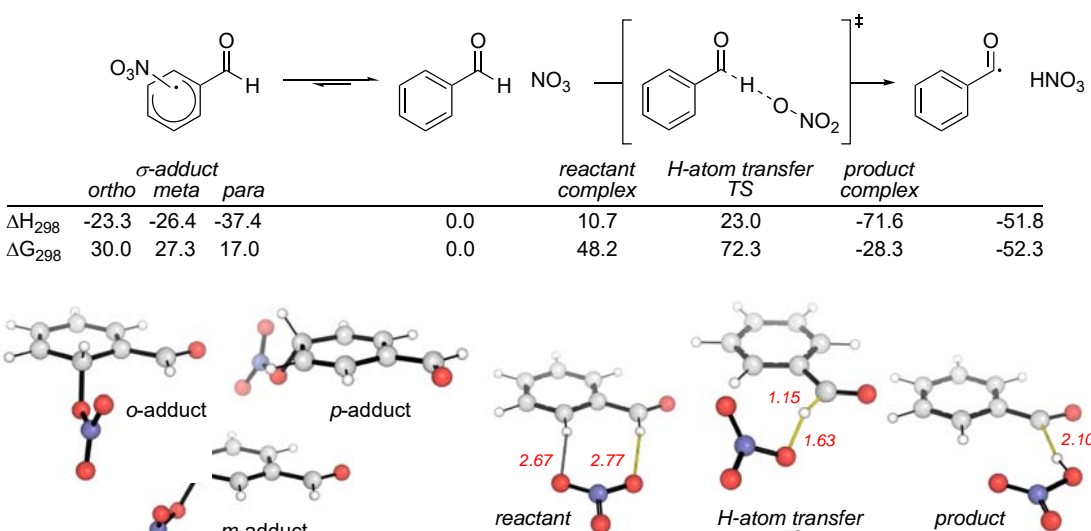

| | σ-adduct | | | reactant complex | H-atom transfer TS | product complex | |
|---|---|---|---|---|---|---|---|
| | ortho | meta | para | | | | |
| $\Delta H_{298}$ | -23.3 | -26.4 | -37.4 | 0.0 | 10.7 | 23.0 | -71.6 | -51.8 |
| $\Delta G_{298}$ | 30.0 | 27.3 | 17.0 | 0.0 | 48.2 | 72.3 | -28.3 | -52.3 |

**Figure 6.** Pathways for the reaction between BA and NO$_3$ calculated using DLPNO-CCSD(T)/cc-pVTZ//BH&HLYP/6-311G(d,p). Enthalpic and Gibbs energy values are in kJ/mol. Key distances in Å.

The computed reaction pathways show that the NO$_3$ radical can form a dearomatized σ-adduct (Figure 6, LHS). The most stable of these adducts (by more than 10 kJ mol$^{-1}$) occurs at the *para*-position. While this complexation is exothermic, it is endergonic by 17.0 kJ mol$^{-1}$ due to unfavorable entropic effects, implying this is a readily reversible process. H-atom transfer first forms a pre-reaction complex with the aldehydic group. This complex can undergo H-atom abstraction to yield the stable products observed in our experiments. Application of the Bigeleisen-Meyer equation to this H-atom transfer transition structure (TS) results in a computationally predicted primary KIE of 2.16 (2.17 with Bell's 1D-tunneling correction) (Bigeleisen and Mayer, 1947;Paton, 2016;Rzepa, 2015). This value is almost identical to the measure KIE of 1.92.

Based on this evidence, we suggest that the reactions of NO$_3$ with aromatic aldehydes lead to the abstraction of the aldehydic H-atom. The cleavage of the aldehydic C-H bond in this step is consistent with the observation that a weaker BDE value is correlated with a larger reaction rate of Reaction (1). CBS-QB3 calculations (SI) imply 2,4-DMBA and 2,5-DMBA, like O-TA, have weaker aldehydic C-H bonds than the other aromatic aldehydes that lack an ortho-substituent. Since the geometry of the formyl radical is more linear than the aldehyde (the C-C-O angle increases by around 4° upon H-atom abstraction), steric strain relief is likely a contributing factor to the C-H bond weakening by an ortho-methyl substituent. Additionally,

charge transfer of 0.22e from BA to $NO_3$ occurs in the computed H-atom transfer TS, consistent with the observation that additional electron-donating substituents, such as methyl groups, promote Reaction (1). To sum up, the combination of the changes in the C-H bond energies and the changes in the variations in the initial addition to form a pre-reaction complex contributed to the observed reactivity trend.

There are potential future experiments that could shed light on the proposed reaction pathway. They include: (1) measurement of the temperature dependence of the reaction rate coefficient; (2) investigating the influence of various isotopic substitutions (e.g., OD reaction studies); (3) studying further substitution of the aromatic ring, for example, with fluorine; and (4) directly detecting the radical formed in the reaction. Further quantum calculations may also be useful.

## 4. Atmospheric Implications

Once emitted from biomass burning and incomplete fuel burning to the atmosphere, the studied aromatic aldehydes would be removed through photolysis and reactions with reactive species such as OH, $NO_3$, and chlorine atoms. The atmospheric lifetimes of the aromatic aldehydes studied in this work have been calculated with respect to the $NO_3$ radical reactions using the rate coefficients, $k_{aro}$, obtained from this work at ambient temperature and pressure, in combination with estimated ambient tropospheric $NO_3$ concentrations, $[NO_3] = 5 \times 10^8$ molecules $cm^{-3}$,(Atkinson, 1991) following the equation: $\tau_{NO3} = 1/(k_{VOC} \times [NO_3])$. It has been pointed out (Brown and Stutz, 2012) that the $NO_3$ radical concentration is highly variable, and we use this value to illustrate the relative loss rates. The calculated lifetimes are shown in Table 3, and they are meant only to compare the loss rates roughly. Actual lifetime needs to be calculated for the location and its condition; it can vary by orders of magnitude from those noted in the table. Based on our measurements, we expect that the aromatic aldehydes' atmospheric lifetimes with respect to $NO_3$ are 37-218 hours (for the assumed $NO_3$ concentrations). Table 3 also presents the lifetime of these seven aromatic aldehydes with respect to OH radicals of $1 \times 10^6$ molecules $cm^{-3}$ (again a rough value characteristic of the mid-tropospheric tropical regions) and Cl atoms (Wingenter et al., 1996) of $1 \times 10^4$ atoms $cm^{-3}$, and the rate coefficients taken from Calvert et al. (2011). It is clear that photolysis dominates, and OH radicals contribute more than Cl atoms to oxidation the aromatic aldehyde in the atmosphere during the daytime. Though $NO_3$ reactions contributed much less on a diurnally averaged basis, they are the only pathways for the degradation of these aldehydes at night. The $NO_3$ reactions could contribute significantly to removing aromatic aldehydes at night in polluted areas with high NOx. Various studies have seen large abundances of the $NO_3$ radicals. For example, the following $NO_3$ mixing ratios have been reported: Pitts et al. Pitts et al., (1984)

up to 430 pptv in the East Los Angeles basin, including Riverside; Wang et al. (2006) up to 200 pptv in the Phoenix downtown area; Brown et al. (2009;2011) up to 400 pptv in Houston urban area; (Platt et al., 1981) around 280 pptv at Deuselbach, Germany; and Asaf et al. (2009;2010) up to 800 around Jerusalem.) In addition, the $NO_3$ reactions also could lead to PAN-type compounds at night.

600

**Table 3.** Summary of rate coefficients and estimated atmospheric lifetimes of benzaldehyde (BA), o-tolualdehyde (O-TA), m-tolualdehyde (M-TA), p-tolualdehyde (P-TA), 2,4-dimethylbenzaldehyde (2,4-DMBA), 2,5-dimethylbenzaldehyde (2,5-DMBA), and 3,5-dimethylbenzaldehyde (3,5-DMBA) with respect to their reactions with OH, $NO_3$, and Cl at $298 \pm 2$ K and atmospheric pressure [a] For comparison, the rough photolysis lifetimes are also noted for roughly 40N during summer.

| | Rate coefficients ($cm^3$ molecule$^{-1}$ s$^{-1}$) | | | Lifetime (hours) | | | |
|---|---|---|---|---|---|---|---|
| | $k_{OH}$[b] | $k_{NO_3}$[c] | $k_{Cl}$[b] | $\tau_{OH}$ | $\tau_{NO_3}$ | $\tau_{Cl}$ | $\tau_{h\nu}$ |
| BA | $1.26\times10^{-11}$ | $2.6\times10^{-15}$ | $9.90\times10^{-11}$ | 2.2-22 | 217 | 281 | ~2 |
| O-TA | $1.89\times10^{-11}$ | $8.7\times10^{-15}$ | $1.86\times10^{-10}$ | 1.5-15 | 64 | 149 | ~2 |
| M-TA | $1.68\times10^{-11}$ | $4.9\times10^{-15}$ | $1.71\times10^{-10}$ | 1.7-17 | 114 | 162 | ~2 |
| P-TA | $1.68\times10^{-11}$ | $4.9\times10^{-15}$ | $1.41\times10^{-10}$ | 1.7-17 | 113 | 197 | ~2 |
| 2,4-DMBA | $3.12\times10^{-11}$ | $1.51\times10^{-14}$ | $8.70\times10^{-11}$ | 0.9-9 | 37 | 319 | <2 |
| 2,5-DMBA | $3.15\times10^{-11}$ | $1.28\times10^{-14}$ | $9.30\times10^{-11}$ | 0.9-9 | 43 | 299 | <2 |
| 3,5-DMBA | $2.78\times10^{-11}$ | $6.2\times10^{-15}$ | $9.30\times10^{-11}$ | 1-10 | 89 | 299 | <2 |

[a] We show a range of lifetimes by assuming [OH] = 1-10 × $10^6$ molecules $cm^{-3}$. [$NO_3$]=5 × $10^8$ molecules $cm^{-3}$ (Atkinson et al., 1991), and [Cl] = 1 × $10^4$ (Wingenter et al., 1996). [b] The rate coefficients for the OH and Cl atom reactions are from Calvert et al. (2011). [c] recommended values in Table 2.

This work also found that the aromatic PAN-type compound, for example, was the main product formed from the reaction of aromatic aldehydes with $NO_3$ radical. The formation of such compounds enables the transport and release of NOx to the remote troposphere, leading to the production of $O_3$. Such a situation may occur in wildland fire plumes.

**Data availability**

The compiled datasets used to produce each figure within this paper are available as Igor Pro files upon request.

**Supplement**

The supplement related to this article is available online at:

**Author contributions**

YR and ARR wrote the paper with input from all authors. YR and LZ conducted the experiments and analyzed the data, MM helped with the data analysis. MI, VD, and SSB were responsible for the CRDS instrument. BR and RSP made the theoretical calculations. AM and ARR designed the experiments and led the study. All coauthors commented on the paper.


**Competing interests**

The authors declare that they have no conflict of interest.

**Acknowledgments**

We are grateful to Timothy Wallington for the IR cross-sections of BAPAN.

**Financial support.** This work is supported by the European Union's Horizon 2020 research and innovation programme through the EUROCHAMP-2020 Infrastructure Activity under grant agreement No. 730997, Labex Voltaire (ANR-10-LABX-100-01), ANR (SEA_M project, ANR-16-CE01-0013, program ANR-RGC 2016), and National Natural Science Foundation

of China (21976106). Work of BR was supported by the U.S. Department of Energy, Office of Science, Office of Basic Energy Sciences, Division of Chemical Sciences, Geosciences, and Biosciences through the Gas-Phase Chemical Physics Program under contract No. DE-AC02-06CH11357. RSP acknowledges the use of the RMACC Summit supercomputer, which is supported by the National Science Foundation (ACI-1532235 and ACI-1532236), the

University of Colorado Boulder and Colorado State University, and the Extreme Science and Engineering Discovery Environment (XSEDE) through allocation TG-CHE180056. Le Studium and CSU supported the work of ARR.

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
