# Peer review of "Reactions of $\text{NO}_3$ with Aromatic Aldehydes: Gas-Phase Kinetics and Insights into the Mechanism of the Reaction"

_Atmospheric Chemistry and Physics, 2021_

## Author Response (AR1)

The authors thank the Reviewers for the comments and suggestions. We have revised our manuscript in response to the reviewers' suggestions and comments. All the changes and responses to the reviewers' comments are listed below, point-by-point, in blue according to the new line numbers in the revised manuscript. The major changes are highlighted in red in the revised manuscript.

**Referee #1**

The authors report rate coefficients for reaction of $NO_3$ radicals with a series of aromatic aldehydes (benzaldehyde, and mono- and di-methyl substituted derivatives), some of which have not been previously studied. Overall, I think this is an excellent study. Multiple approaches (relative and absolute) are used to obtain the kinetic data – the overall good agreement between the methods lends confidence to the obtained values. The work is very nicely put into broader context through a product study (showing formation of a PAN species), a determination of the H/D KIE for benzaldehyde, as well as theoretical calculations, etc., all of which contribute to a deeper understanding of the mechanism involved in the reactions and some of the reasons for the variation in the rate coefficients for the different methyl substitutions. The work is, in my opinion, publishable subject to minor revisions. I have one concern of some potential significance (first point below), as well as a few questions and suggestions which the authors may wish to consider.

We thank the reviewer for the positive comments.

I am not sure what the connection is between panel (a) and the other panels in Figure 2. They have different timescales, and different concentrations. Is there something mislabeled here? Or am I missing something?

Responses: We have revised the time scale in panel (a) to be in seconds and the concentration scales to be in ppbv. Just to clarify: The panels b()-(d) are in linear scale so that the fit can be compared with the data.

Abstract, Line 36: It might be better to say $NO_2$, rather than NOx, as NO would not lead to PAN formation.

Responses: Agreed. $NO_x$ is changed to "$NO_2$".

Introduction, Line 44: Maybe pyrogenic sources is a better description than biogenic? None of the sources listed are what I would consider 'biogenic'.

Responses: Thank you. Indeed it is better terminology. We changed "biogenic" to "pyrogenic".

Line 175: 'recommended by Burkholder et al…' might sound better.

Responses: Thank you. Done!

Line 205: There is at least one other measurement of NO3 + methyl methacrylate from Canosa-Mas et al. (1999). Is it best to include this (slightly higher) value in arriving at the best value for the reference rate?

Responses: We agree with the reviewer that the rate constant of $NO_3$ reaction with methyl methacrylate has been determined by Canosa-Mas et al., (1999), Wang et al., (2010) and Sagrario Salgado et al., (2011). The reported experiments were very similar to that of Zhou et al., (2017) where the rate coefficient of interest was measured by both absolute and Relative methods. To enable placing the measured relative rate coefficient comparable to our absolute method, we prefer to use Zhou et al. value. We have provided all the measured values so that one could recalculate the rate coefficient for any value for the reference reactions.

Line 369: Wouldn't increasing (not decreasing) the NO2 concentration suppress pathway B (by favoring PAN formation in pathway A)?

Responses: Apologies. That is correct. It was an error. It is now fixed.

Around line 485: Regarding the bond strength calculations, the para-ta and meta-ta have similar bond strengths to benzaldehyde, yet faster rate coefficients. Are the authors stating that this is due to the electron-donating effect of the methyl group, while the even faster o-ta rate coefficient includes an additional effect due to the weaker bond strength? Maybe a summary statement or two would be helpful at the end of this section? Also, is there any connection that can be made between the results here and the observation that k(NO3/aliphatic aldehydes) increase (to a point) with size of the alkyl chain (e.g., Noda et al., 2003)?

Responses: Yes, we are saying that the combination of the pre-reaction complex formation and the changes in the C-H bond energies contribute to the observed trends. We have added a summing up statement: "To sum up, the combination of

the changes in the C-H bond energies and the changes in the variations in the initial addition to form a pre-reaction complex contributed to the observed reactivity trend."

Thank you for pointing us to the work of Noda et al. We hesitate to comment on the variation of the reactivity of $NO_3$ with alkane chain length in aliphatic aldehydes without doing further calculations since strengths of some of the secondary C-H bond strengths and their number, as well as steric effects could influence the reactivity.

**Referee #2**

This manuscript presents the results of experimentally determined rate constants for the reaction of $NO_3$ with a series of aromatic aldehydes. The authors employ two methods of determining the rate constants. Results from the absolute and relative methods are in good agreement, lending weight to the authors' overall findings. The authors first compare results for systems with published rate constant values, and then extend the study to three new systems. Support for the suggested reaction mechanisms using isotope effects and BDEs are convincing. Overall, this study is well-conducted and well-presented. The manuscript is publishable after considering the minor revisions listed below.

We thank you reviewer for the positive comments.

Specific Comments

The absolute method of determining the rate constants was repeated four times for each system. The relative method was only duplicated. Another experiment repeating the relative method (n=3) would be preferred.

Responses: Thank you. We repeated the relative rate coefficient measurements. The reported rate coefficients did not change noticeably but our uncertainties decreased. Now, the average of three measurements are reported in Table S2.

Line 130: For step one determining the dilution coefficient, were the $SF_6$ and air continually added to mimic the continual addition of air with $N_2O_5$ during the experiment? As written, the initial 30 minute observation would seem to determine diffusion. Some rewording could be useful here.

Responses: Yes, we continually added air to make up for the withdrawal of air for analyses. We corrected the sentence to read "$SF_6$ was added at the beginning of the experiment and monitored to measure the dilution rate coefficient. During this time, air was continually added to make up for the loss due to the withdrawal for analysis using CIMS and CRDS".

Line 168: Do the authors mean the residuals were minimized?

Responses: Thank you. Yes. To be clear, we have rewritten the sentence to read: "In the algorithm, the sum of squares of the difference between calculated and measured values of both the $NO_3$ and $N_2O_5$ were minimized while varying the input parameters"

Figure 2: Panel b does not significantly add to understanding the system and might be removed.

Responses: Panel B is: Temporal profiles in linear scale and the fits to the data as discussed in the text to determine the wall loss of $N_2O_5$ and $NO_3$ during the experiment. We prefer to retain this panel because it shows the goodness of fit of the data. (That difference would not be visible on a log scale and so we prefer to show it on a linear scale. We have also changes the axes on panel (a) to be consistent with other panels.

Line 298-299: The overlap of combined error bars is not a valid statistical test or comparison. I recommend removing this statement.

Responses: Thank you. We just want to note that the overlap statement was based on the errors based on including estimated systematic errors. That was not a statistical test. In any case, we agree with the reviewer's sense that the sentence was redundant because the previous sentences give the sense of the agreement.

Section 3.1.4: Table S5, which compares this work with previously published experimental values, contains key findings that are discussed in detail in this section. This data is also presented in graphical form in Figure 5 later in the manuscript. The reader would be assisted if immediate access to these data were presented in this section. Solutions could include the following options: the text could referred to both the figure and the table to eliminate continual flipping between the manuscript and the SI, or Table S5 could be relocated to the main text of the manuscript, or combine Table 1 with Table S5. It is also unclear why the authors have selected a different exponent for Figure 5 than what is used in the rest of the manuscript.

Responses: Thanks the reviewer for the suggestion, We referred Table S5 (Now as the Table 2 in the main text) in the caption of Figure 5 as "values are presented in Table 2". We also mention it in section on the results and discussion.

Figure 3: The caption is missing a description of the different colors/markers used in the figure. Could the authors please clarify.

Responses: We added the following note in the caption of Figure 3:

"Note: the different colors/markers presented the repeated experiments with different initial $NO_2$ concentration as: red filled cycles: initial $[NO_2]$= 1.4-2.9 $\times10^{12}$ molecule $cm^{-3}$, blue triangle: initial $[NO_2]$= ~2.7 $\times10^{13}$ molecule $cm^{-3}$, green square : $[NO_2]_0$= ~1.4$\times10^{14}$ molecule $cm^{-3}$ and red open diamond: $[NO_2]_0$= ~3.3 $\times10^{14}$ molecule $cm^{-3}$."

Figure S1 (and Table S1): Please include the offset values in the caption as you have done with previous figures. Would it make sense to also list the $R^2$ values in Table S1? I am assuming the detection sensitivity listed in this table is the slope. It would also be useful to have the uncertainty listed here defined.

Responses: For Figure S1, we added the following sentence in the figure caption: "For the sake of clarity, the plots of 2,4-DMBA, 2,5-DMBA, 3,5-DMBA, M-TA and P-TA are displaced vertically by 0.05, 0.1, 0.15, 0.2 and 0.2, respectively.".
We also added the $R^2$ (0.95-0.99) and defined the uncertainty (2 times of standard deviation of the linear fitting in Figure S1) in the Table S1 which was asked by the reviewer.

Technical Comments

Line 51: "their" is ambiguous in this sentence structure. I would recommend switching the placement of "their" and "aromatic aldehydes" to increase clarity.

Responses: Thank you. This sentence has been rewritten.

**Referee #3**

This manuscript describes a detailed investigation of the reactions of seven aromatic aldehydes with the nitrate radical. Rate coefficients have been determined using two experimental methods and the results are in very good agreement. Theoretical calculations have also been carried out to understand the reactivity patterns and mechanisms. The data obtained in this work is of high quality and interpreted well. Overall, the manuscript is well written and presented. I recommend publication after the authors satisfactorily address my minor comments below.

We thank the reviewer for these positive comments.

1. My main comment is that the authors have neglected direct photolysis as a potential degradation pathway for the aromatic aldehydes. As demonstrated by Clifford et al. (2011), this pathway is certainly im portant for o-tolualdehyde, where the lifetime due to photolysis can be as short as 1-2 hours. The authors should incorporate this into the relevant parts of their manuscript (lines 51-55, lines 543-560, Table 2).

Responses: We agree and are grateful to the reviewer for pointing out this obvious oversight. We apologize for leaving photolysis out. We were so focused on comparing $NO_3$ reactive loss with other free radical reactions that we neglected to note the importance (dominance) of photolysis during daytime. We have modified the introduction to note:
"It is known that the tropospheric removal of aromatic aldehydes occurs through photolysis (Clifford et al., 2011; Thiault et al.2004) and theirs reactions with the OH radical. During daytime, photolysis and reaction with OH dominates their atmospheric loss"
In addition, we added a column to Table 2 to show the rough photolysis rate, which, of course, vary with time, location and season. We did not find good UV spectra of the three DMBA, which surely would absorb as well as or more than the mono-substituted aldehydes. Therefore we note their photolytic lifetime to be less than 2 hours.
We have re-written the atmospheric introduction section to take these changes into account.
In addition, we have included the lifetime due to photolysis in the Table 2. The footnote now notes the variability of these loss pathways.

2. Line 103: please add detection limit for nitrate radical in units of molecule/cm$^3$.

Responses: we added the detection limit for $NO_3$ radical as $1.23\times10^7$ molecule cm$^{-3}$ in line 106.

3.  Line 126: replace "watching" with "monitoring".

Responses: Done in line 130.

4.  Line 143: should be [ref]

Responses: we changed the [reference] to [ref] in line 146.

5.  Lines 174, 177 and maybe elsewhere: use a big K for the equilibrium constant.

Responses: We correct "$k_{eq}$" to "$K_{eq}$".

6.  Table 1, Table 2 and tables in SI: use "rate coefficients" instead of "rate constants" to be consistent with other parts of the manuscript.

Responses: Done

Reference for Clifford et al:
G.M. Clifford, A. Hadj-Aïssa, R.M. Healy, A. Mellouki, A. Muñoz, K. Wirtz, M. Martín
Reviejo, E. Borrás, J.C. Wenger
The atmospheric photolysis of o-tolualdehyde
Environ. Sci. Technol., 40 (2011), pp. 9649-9657

---

## Author Response (AR2)

The authors thank the Editors for the comments and suggestions. We have revised our manuscript in response to the editor's suggestions and comments. All the changes and responses to the editor's comments are listed below, point-by-point in blue. The changes are highlighted in red in the revised manuscript.

Comments to the Author:

Dear Authors,

Thanks for carefully addressing the reviewer comments. I'm almost ready to accept this for full publication in ACP. However, I noted a few minor edits that need to be made before I can fully accept this. Once you address these, I'll be sure to quickly move in reviewing them and making a final decision. Most sincerely, Jason Surratt

Minor Comments:

1.) Line 58-60: Please add citations to the published literature that aromatic aldehydes can be degraded at night to yield SOA or have consequences for $O_3$.

Responses:

We had noted that the reaction can have consequences to SOA formation. In light of the possible misunderstanding, we have now changed the text to read: To the best of our knowledge, there are no studies that have measured the formation of secondary organic aerosol (SAO) from the title reaction in either a chamber or in the atmosphere. One could suspect that aromatic aldehydes may be degraded at night with consequences for ozone and secondary organic aerosol (SOA) formation.

2.) Line 585-586: please delete ", as pointed out by" as you don't need this wording here.

Responses: Done.

3.) Line 597: Can the authors provide more of a justification of the conclusion that "The $NO_3$ reactions can contribute significantly to removing aromatic aldehydes at night in a polluted area with high NOx?" How do we know this? Is there literature you can cite that would suggest what levels of $NO_3$ would exist in these high-NOx polluted areas to make $NO_3$ reactions with these aromatic aldehydes more substantial at night?

Responses: Thank you. Yes, there are many studies in the urban regions that have measured $NO_3$ levels approaching 0.5 ppbv. We have changed the text as follows:

The $NO_3$ reactions could contribute significantly to removing aromatic aldehydes at night in polluted areas with high NOx. Various studies have seen large abundances of the $NO_3$ radicals. For example, the following $NO_3$ mixing ratios have been reported: Pitts et al. Pitts et al., (1984) up to 430 pptv in the East Los Angeles basin, including Riverside; Wang et al. (2006) up to 200 pptv in the Phoenix downtown area; Brown et al. (2009;2011) up to 400 pptv in Houston urban area; (Platt et al., 1981) around 280 pptv at Deuselbach, Germany; and Asaf et al. (2009;2010) up to 800 around Jerusalem.) In addition, the $NO_3$ reactions also could lead to PAN-type compounds at night.

---

## Author Response (AR3)

The authors thank the Editors for the comments and suggestions. We have revised our manuscript in response to the editor's suggestions. The changes and responses to the editor's comments are listed below, point-by-point in blue. The changes are highlighted in red in the revised manuscript.

Comments to the Author:

Dear Authors, thanks for making the suggested changes. I noted in your new revisions a couple of minor mistakes on lines 58-62:

Please change "secondary organic aerosol (SAO)" to "secondary organic aerosol (SOA)"

Response: Done

Also, on line 61, please change secondary organic aerosol (SOA)" to "SOA" as you already defined the acronym in the prior sentence.

Response: Done